# Diarrhea and related personal characteristics among Japanese university students studying abroad in intermediate- and low-risk countries

Michiyo Yamakawa[1,2]*, Toshihide Tsuda[3], Keiko Wada[1], Chisato Nagata[1], Etsuji Suzuki[2,4]

1 Department of Epidemiology and Preventive Medicine, Graduate School of Medicine, Gifu University, Gifu, Japan, 2 Department of Epidemiology, Graduate School of Medicine, Dentistry and Pharmaceutical Sciences, Okayama University, Okayama, Japan, 3 Department of Human Ecology, Graduate School of Environmental and Life Science, Okayama University, Okayama, Japan, 4 Department of Epidemiology, Harvard T.H. Chan School of Public Health, Boston, Massachusetts, United States of America

* myamak@gifu-u.ac.jp

**Data Availability Statement:** Data cannot be shared publicly because of ethical restrictions and legal framework of Japan. It is prohibited by the Act on the Protection of Personal Information (Act No.

## Abstract

Despite an increasing number of students studying abroad worldwide, evidence about health risks while they are abroad is limited. Diarrhea is considered the most common travelers' illness, which would also apply to students studying abroad. We examined diarrhea and related personal characteristics among Japanese students studying abroad. Japanese university students who participated in short-term study abroad programs between summer 2016 and spring 2018 were targeted (n = 825, 6–38 travel days). Based on a 2-week-risk of diarrhea (passing three or more loose or liquid stools per day) among travelers by country, the destination was separated into intermediate- and low-risk countries. After this stratification, the associations between personal characteristics and diarrhea during the first two weeks of their stay were evaluated using logistic regression models. Among participants in intermediate-risk countries, teenagers, males and those with overseas travel experience were associated with an elevated risk of diarrhea; the odds ratios (95% confidence intervals) were 2.42 (1.08–5.43) for teenagers (vs. twenties), 1.93 (1.08–3.45) for males (vs. females) and 2.37 (1.29–4.33) for those with overseas experience (vs. none). Even restricting an outcome to diarrhea during the first week did not change the results substantially. The same tendency was not observed for those in the low-risk countries. Teenage students, males and those with overseas travel experience should be cautious about diarrhea while studying abroad, specifically in intermediate-risk countries.

## Introduction

Economic and social globalization has led to a growing number of students studying abroad worldwide [1]. In Japan, the government has strongly promoted overseas study for the younger generation to improve their international understanding and foreign language skills [2],

57 of 30 May 2003, amendment on 9 September 2015) to publicly deposit the data containing personal information. Ethical Guidelines for Medical and Health Research Involving Human Subjects enforced by the Ministry of Education, Culture, Sports, Science and Technology and the Ministry of Health, Labour and Welfare, Japan also restricts the open sharing of the epidemiologic data. Data are however available on request from the Okayama University Graduate School of Medicine, Dentistry and Pharmaceutical Sciences and Okayama University Hospital, Ethics Committee (mae6605@adm.okayama-u.ac.jp) for researchers who meet the criteria for access to data.

**Funding:** This work was supported by JSPS KAKENHI [Grant Number JP17K02111]. There was no additional external funding received for this study. The funder had no role in study design, data collection and analysis, decision to publish, or preparation of the manuscript.

**Competing interests:** The authors have declared that no competing interests exist.

leading to an increase in Japanese students studying abroad in recent years. Although the USA has been the most popular country for overseas studies among Japanese university students; Asia is now the most popular region (e.g., China, South Korea, Philippines and Thailand). The number of students going to Asian countries for studies increased from approximately 10,000 (24% of all overseas study destinations) in 2010 to 45,000 (39%) in 2018 [3].

While abroad, students need to be aware of the dangers of developing health problems in an unfamiliar living environment [4]. Among them, diarrhea is the most common problem among travelers moving from developed to tropical and developing regions [5, 6]. This may also apply to students studying abroad, although past studies mostly targeted tourists [7–9]. Based on the data about travelers' diarrhea risk during the first two weeks of stay, destinations have been categorized into three groups: low risk (<8%), intermediate risk (≥8 to <20%), and high risk (≥20%) [5].

In general, travelers' personal characteristics (e.g., age, country of origin, immunity, and low-gastric acidity) were shown to be associated with a higher risk of diarrhea [5, 6]. However, despite an increasing number of students studying abroad, evidence about their personal characteristics associated with diarrhea remains sparse. A past study reported that healthcare majors and travel to developing regions were associated with an elevated risk of diarrhea among Swedish students studying abroad, although healthcare students were more likely to travel to developing regions than non-healthcare students [8]. We thus aimed to examine the association of personal characteristics, including study major, with diarrhea among Japanese university students studying abroad according to the destination's risk categories.

## Materials and methods

### Study participants

The participants were Japanese students of Okayama University in Okayama, Japan, who traveled abroad for short-term study abroad programs during vacations in summer (August to September) and spring (February to March) from April 2016 to March 2018 (n = 825). The programs, including various experience-based activities, were conducted primarily for the purpose of improving cross-cultural understanding and foreign language skills. The destinations were Australia, Canada, China, France, Ireland, Malaysia, Singapore, UK, South Korea, Thailand, and USA (mainland and Guam). The travel period, depending on the program, ranged from six days for a program in Singapore to 38 days for a program in Australia. All participants were required to attend a lecture on health problems related to overseas travel and precautions, given by the university two months before their departure. In the present study, two surveys were conducted using a paper-based questionnaire at the lecture (pre-travel survey) and a web-based questionnaire 1–2 months after return to Japan (post-travel survey). Invitations to the web survey were made first for all participants by email and second for non-respondents to the first invitation at briefing sessions for the programs. In the pre-travel survey, 801 participants responded to the questionnaires (97.1% response rate). Among them, 126 were lost to follow-up at the post-travel survey and two were excluded because of missing information about the first diarrheal episode while abroad. Finally, 673 participants were analyzed (84.0% response rate).

### Data collection

The pre-travel survey contained questions about programs including destinations, travel period, and personal characteristics including age, grade, time of participation, academic year, sex, study major, underlying medical conditions (no or yes), ABO blood group, and overseas travel experience. Considering the distribution of participants' age (mean 19, min 18 to max

28 years old), we dichotomized age group as teenagers or twenties. We also dichotomized grade as 1st or $\geq$ 2nd year and study major as healthcare or non-healthcare. The "healthcare" category included participants who majored in medicine, dentistry, pharmacy, nursing, and other allied health professions. ABO blood group and overseas travel experience were considered indicators of preexisting immunity to the agents causing diarrhea [6, 10–13]. Considering the immunity against norovirus [14] and *Vibrio cholerae* [15], ABO blood group was dichotomized as 'O' or other types including unknown blood type. Overseas travel experience was categorized into three groups; "no," "yes, but never visited the current destination," and "yes, and have visited the current destination." Based on the definition mentioned in the Introduction [5], the destinations were separated into intermediate-risk countries (China, Malaysia, Singapore, South Korea, and Thailand) and low-risk countries (Australia, Canada, France, Ireland, UK, and USA). Considering the equatorial climate, we included Singapore as intermediate-risk countries. The post-travel survey contained questions about the first diarrheal episode while abroad, including the onset date, the frequency of bowel movement, symptoms, and coping strategies during the episode. We also asked about any diarrheal episodes after return to Japan, including the onset date.

Following the criteria of the World Health Organization, we defined diarrhea as the passage of three or more loose or liquid stools per day [16, 17]. Considering that diarrhea risk varies by travel period, and most cases occur within the first two weeks [18], the first diarrheal episode during the first two weeks was used as an outcome. Those who returned to Japan without a diarrheal episode within two weeks were treated as censored.

This study was approved by the Okayama University Graduate School of Medicine, Dentistry and Pharmaceutical Sciences and Okayama University Hospital, Ethics Committee (No. R1606-039). Written informed consent was obtained from all participants before the first survey. Study and participation details were described on the written documents and the institution's official website, so that parents can refuse to let their teenage children participate in the study even after those children agreed to participate.

## Analysis

All analyses were stratified by the destination's risk categories (intermediate- or low-risk countries). We first examined the distributions of personal characteristics, the first 2-week risk of diarrhea and its characteristics. Then, we used logistic regression models to estimate crude odds ratios (ORs) and 95% confidence intervals (CIs) for the associations between personal characteristics and diarrhea during the first two weeks using the following reference categories: twenties, 2nd year or higher, summer in Japan, 2016 academic year, females, non-healthcare major, no underlying medical conditions, other blood types, and no overseas travel experience. Then, we estimated ORs and 95% CIs for the associations simultaneously adjusting for the covariates with a two-sided *P* value of less than 0.10 in each crude model. Finally, we conducted three sensitivity analyses. First, regarding the censored subjects because of their return to Japan or no diarrheal episodes within the first two weeks, we extended the "at-risk" period by two days after they were censored. For this analysis, the 2-day incubation period of enterotoxigenic *Escherichia coli* was considered as a presumed pathogen of diarrhea. Second, we reanalyzed the data by using the first episode during the first week as an outcome. Third, we excluded Singapore from the category of intermediate-risk countries, and repeated the analyses.

A *P* value of less than 0.05 (two-sided test) was considered significant. Stata SE, version 14.2, statistical software (StataCorp., College Station, TX, USA) was used for all analyses. Stata SE statistical software (version 16.1; StataCorp, College Station, TX, USA) was used for all the analyses.

## Results

Table 1 shows personal characteristics of Japanese university students participating in the programs. Compared with the participants in low-risk countries, those in intermediate-risk countries traveled for a shorter period, and were more likely to be freshmen and studying abroad during their summer vacations. Among those with underlying medical conditions, one participant in an intermediate-risk country had inflammatory bowel disease and another in a low-

**Table 1. Personal characteristics and the first 2-week risks of diarrhea among Japanese university students participating in short-term study abroad programs in intermediate- and low-risk countries[a].**

| | Intermediate-risk countries [a] (n = 347) | Low-risk countries [a] (n = 326) |
|---|---|---|
| **Travel period, median days (Q1, Q3)** | 7 (6, 15) | 30 (27, 31) |
| **Age group, n (%)** | | |
| 10s | 268 (77.2) | 212 (65.0) |
| 20s | 79 (22.8) | 114 (35.0) |
| **Grade, n (%)** | | |
| 1st | 233 (67.2) | 156 (47.9) |
| 2nd or higher | 114 (32.9) | 170 (52.2) |
| **Academic year, n (%)** | | |
| 2016 | 110 (31.7) | 118 (36.2) |
| 2017 | 101 (29.1) | 109 (33.4) |
| 2018 | 136 (39.2) | 99 (30.4) |
| **Time of participation, n (%)** | | |
| Summer | 226 (65.1) | 134 (41.1) |
| Spring | 121 (34.9) | 192 (58.9) |
| **Sex, n (%)** | | |
| Male | 135 (38.9) | 107 (32.8) |
| Female | 212 (61.1) | 219 (67.2) |
| **Study major, n (%)** | | |
| Non-healthcare | 318 (91.6) | 282 (86.5) |
| Healthcare[b] | 29 (8.4) | 44 (13.5) |
| **Underlying medical conditions, n (%)** | | |
| No | 300 (86.5) | 277 (85.0) |
| Yes | 47 (13.5) | 49 (15.0) |
| **ABO blood group** | | |
| Type O | 112 (32.3) | 96 (28.5) |
| Other types[c] | 235 (67.7) | 233 (71.5) |
| **Overseas travel experience, n (%)** | | |
| No | 166 (47.8) | 110 (33.7) |
| Yes, but never visited the destination | 166 (47.8) | 175 (53.7) |
| Yes, and have visited the destination | 15 (4.3) | 41 (12.6) |
| **Total diarrhea cases, n (%, 95% CI)** | 60 (17.3, 13.3–21.3) | 33 (10.1, 6.8–13.4) |
| **2-week diarrhea cases, n (%, 95% CI)** | 60 (17.3, 13.3–21.3) | 29 (8.9, 5.8–12.0) |

Abbreviations: n, number; Q1, 1st quartile; and Q3, 3rd quartile.

[a] Intermediate-risk countries included Malaysia, Singapore, Thailand, and China, and low-risk countries included Australia, USA, France, UK, Ireland, Canada, and South Korea.

[b] The category of "healthcare" included participants who majored in medicine, dentistry, pharmacy, nursing and other allied health professions.

[c] Types A, B and AB, and blood types unknown were included.

risk country had renal disease (data not shown). The first 2-week risks of diarrhea (95% CI) were 17.3% (13.3–21.3) and 8.9% (5.8–12.0) in intermediate- and low-risk countries, respectively, although the travel period for 163 participants in Singapore (47%), which was categorized as intermediate countries, was only six days.

Table 2 shows clinical characteristics of diarrhea cases. In both destination categories, 75% of total cases occurred within 7–8 days of their stay, and 52% had abdominal cramps. Moreover, during the diarrheal episode, approximately half did not employ any coping strategies or took medicines they carried from Japan.

Table 3 shows the association between personal characteristics and diarrhea during the first two weeks. In intermediate-risk countries, having overseas travel experience (irrespective of its destination) was associated with an elevated risk of diarrhea, compared with no overseas travel experience. In low-risk countries, the 2017 academic year was associated with an elevated risk as compared to 2016.

Table 4 shows the results for the ORs simultaneously adjusting for age group, academic year, sex, and overseas travel experience. For the analyses, we reorganized overseas travel experience into two groups, i.e., no or yes, because of similarly elevated ORs for the two original groups of "yes" ("yes, but never visited current destination," "yes, and have visited current destination"). In intermediate-risk countries, age group, sex and overseas travel experience were associated with diarrhea during the first two weeks, compared with each reference category. The adjusted ORs (95% CIs) were 2.42 (1.08–5.43) for 18–19 years, 1.93 (1.08–3.45) for males,

**Table 2.** Clinical characteristics of diarrhea during the first two weeks in intermediate-risk and low-risk countries[a].

|  | Intermediate-risk countries [a] (n = 60) | Low-risk countries[a] (n = 29) |
|---|---|---|
| **First diarrhea onset, median days (Q1, Q3)** | 7 (4, 8) | 3 (1, 7) |
| **Frequency of bowel movement, n (%)** | | |
| 3–4 times/day | 39 (65.0) | 17 (58.6) |
| 5–6 times/day | 11 (18.3) | 8 (27.6) |
| 7–8 times/day | 9 (15.0) | 4 (13.8) |
| 9 times/day | 1 (1.7) | 0 (0) |
| **Symptoms other than loose stools, n (%)** | | |
| Abdominal cramps | 31 (51.7) | 15 (51.7) |
| Watery stools | 27 (45.0) | 8 (27.6) |
| Fever | 6 (10.0) | 1 (3.4) |
| Nausea, vomiting | 5 (8.3) | 6 (20.7) |
| Fecal urgency | 3 (5.0) | 0 (0) |
| Tenesmus | 1 (1.7) | 4 (13.8) |
| **Coping strategies during diarrheal episodes, n (%)** | | |
| None[b] | 28 (46.7) | 10 (34.5) |
| Taking medicines carried from Japan | 26 (43.3) | 15 (51.7) |
| Changing planned activities | 4 (6.7) | 5 (17.2) |
| Consulting local doctors | 4 (6.7) | 1 (3.4) |
| Taking medicines available at local pharmacies | 1 (1.7) | 3 (10.3) |

Abbreviations: n, number; Q1, 1st quartile; and Q3, 3rd quartile.

[a] Intermediate-risk countries included Malaysia, Singapore, Thailand, and China; and low-risk countries included Australia, USA, France, UK, Ireland, Canada, and South Korea.

[b] Two participants without coping strategies during diarrheal episodes were included in intermediate-risk countries.

**Table 3. ORs for diarrhea during the first two weeks associated with personal characteristics among the participants in intermediate- and low-risk countries[a].**

| | Intermediate-risk countries [a] (n = 347) | | | Low-risk countries [a] (n = 326) | | |
|---|---|---|---|---|---|---|
| | Case, n (%) | cOR (95% CI) | P | Case, n (%) | cOR (95% CI) | P |
| **Age group** | | | | | | |
| 10s | 52 (19.4) | 2.14 (0.97, 4.71) | 0.060 | 19 (9.0) | 1.02 (0.46, 2.28) | 0.954 |
| 20s | 8 (10.1) | 1 (ref.) | | 10 (8.8) | 1 (ref.) | |
| **Grade** | | | | | | |
| 1st | 44 (18.9) | 1.43 (0.77, 2.66) | 0.264 | 15 (9.6) | 1.19 (0.55, 2.54) | 0.662 |
| 2nd or higher | 16 (14.0) | 1 (ref.) | | 14 (8.2) | 1 (ref.) | |
| **Academic year** | | | | | | |
| 2016 | 17 (15.5) | 1 (ref.) | | 6 (5.1) | 1 (ref.) | |
| 2017 | 22 (21.8) | 1.52 (0.76, 3.07) | 0.239 | 16 (14.7) | 3.21 (1.21, 8.54) | 0.019 |
| 2018 | 21 (15.4) | 1.00 (0.50, 2.00) | 0.998 | 7 (7.1) | 1.05 (0.37, 2.99) | 0.541 |
| **Time of participation** | | | | | | |
| Summer | 38 (16.8) | 1 (ref.) | | 13 (9.7) | 1 (ref.) | |
| Spring | 22 (18.2) | 1.10 (0.62, 1.96) | 0.748 | 16 (8.3) | 0.85 (0.39, 1.82) | 0.670 |
| **Sex** | | | | | | |
| Male | 30 (22.2) | 1.73 (0.99, 3.04) | 0.054 | 6 (5.6) | 0.50 (0.20, 1.28) | 0.151 |
| Female | 30 (14.2) | 1 (ref.) | | 23 (10.5) | 1 (ref.) | |
| **Study major** | | | | | | |
| Non-healthcare | 56 (17.6) | 1 (ref.) | | 24 (8.5) | 1 (ref.) | |
| Healthcare | 4 (13.8) | 0.75 (0.25, 2.24) | 0.604 | 5 (11.4) | 1.38 (0.50, 3.82) | 0.538 |
| **Underlying medical conditions** | | | | | | |
| No | 55 (18.3) | 1 (ref.) | | 24 (8.7) | 1 (ref.) | |
| Yes | 5 (10.6) | 0.53 (0.20, 1.40) | 0.201 | 5 (10.2) | 1.19 (0.43, 3.31) | 0.727 |
| **ABO blood group** | | | | | | |
| Type O | 20 (17.9) | 1.06 (0.59, 1.91) | 0.847 | 9 (9.7) | 1.14 (0.50, 2.61) | 0.754 |
| Other types [b] | 40 (17.0) | 1 (ref.) | | 20 (8.6) | 1 (ref.) | |
| **Overseas travel experience** | | | | | | |
| No | 19 (11.5) | 1 (ref.) | | 10 (9.1) | 1 (ref.) | |
| Yes, but never visited the destination | 37 (22.3) | 2.22 (1.22, 4.05) | 0.009 | 15 (8.6) | 0.94 (0.41, 2.17) | 0.880 |
| Yes, and have visited the destination | 4 (26.7) | 2.81 (0.81, 9.72) | 0.102 | 4 (9.8) | 1.08 (0.32, 3.66) | 0.900 |

Abbreviations: n, number; cOR, crude odds ratio; ref, reference; CI, confidence interval.

[a] Intermediate-risk countries included Malaysia, Singapore, Thailand, and China; and low-risk countries included Australia, USA, France, UK, Ireland, Canada, and South Korea.

[b] Types A, B and AB, and blood type unknown were included.

and 2.37 (1.29–4.33) for having overseas travel experience. In low-risk countries, the elevated OR for 2017 (3.03, 95% CI: 1.12–8.22) remained significant even after adjusting for the covariates.

In the first sensitivity analysis, when we extended the "at risk" period by two days for those who were censored, the number of cases became 66 and 32 in intermediate- and low-risk countries, respectively; however, the main findings did not change substantially, e.g., OR (95% CIs) of 2.19 (1.23–3.90) for having overseas travel experience in intermediate-risk countries. In the second analysis for 1-week diarrhea risk, the associations for age group, sex, overseas travel experience and academic year showed the same tendencies, although the OR for 2017 in low-risk countries remained significant. In the third analysis excluding Singapore from the category of intermediate-risk countries, the results were similar to those of the main analyses,

**Table 4. Adjusted ORs for diarrhea during the first two weeks associated with age group, academic year, sex, and overseas travel experience among the participants in intermediate- and low-risk countries[a].**

| | Intermediate-risk countries [a] | | Low-risk countries [a] | |
|---|---|---|---|---|
| | aOR [b] (95% CI) | P | aOR [b] (95% CI) | P |
| **Age group** | | | | |
| 10s | 2.42 (1.08, 5.43) | 0.032 | 1.07 (0.47, 2.45) | 0.867 |
| 20s | 1 (ref.) | | 1 (ref.) | |
| **Academic year** | | | | |
| 2016 | 1 (ref.) | | 1 (ref.) | |
| 2017 | 1.40 (0.68, 2.88) | 0.364 | 3.03 (1.12, 8.22) | 0.030 |
| 2018 | 1.00 (0.49, 2.05) | 0.995 | 1.36 (0.43, 4.29) | 0.594 |
| **Sex** | | | | |
| Male | 1.93 (1.08, 3.45) | 0.027 | 0.58 (0.22, 1.49) | 0.257 |
| Female | 1 (ref.) | | 1 (ref.) | |
| **Overseas travel experience** | | | | |
| No | 1 (ref.) | | 1 (ref.) | |
| Yes [c] | 2.37 (1.29, 4.33) | 0.005 | 0.89 (0.39, 2.03) | 0.787 |

Abbreviations: aOR, adjusted odds ratio; CI, confidence interval; ref, reference.

[a] Intermediate-risk countries included Malaysia, Singapore, Thailand, and China; and low-risk countries included Australia, USA, France, UK, Ireland, Canada, and South Korea.

[b] Age group, academic year, sex, and overseas travel experience were simultaneously adjusted for.

[c] Two overseas travel experience categories (ever or never visited the destination) were combined.

although statistical power was reduced, e.g., OR (95% CIs) of 2.95 (0.90–4.21) for having overseas travel experience (S1 Table).

## Discussion

In the present study, we examined diarrhea and related personal characteristics among Japanese university students participating in short-term study abroad programs in intermediate- and low-risk countries. Teenage students, males and those with overseas experience appeared to be associated with an elevated risk of diarrhea in intermediate-risk countries. The high response rates at the pre- and post-travel surveys (97.1% and 84.0%, respectively) will strengthen the validity of these findings.

The proportion of travelers with diarrhea who changed planned activities varies by country [5], e.g., 11.3–20.6% in Thailand [19–22], 11.9% in Brazil and 45.7% in India [12]. The present study showed that more cases in low-risk countries (17.2%) tended to change planned activities than those in intermediate-risk countries (6.7%), although diarrhea appeared to be mild and self-limited. This could be due to higher frequencies of accompanying symptoms among those in low-risk countries (e.g., nausea or vomiting and tenesmus). Another possible reason is that a longer period of stay made it easier to adjust their plans. In addition, approximately half took medicines they carried from Japan during the diarrhea episode. We previously reported that 10% carried antibiotics among the same study population [23], and thus some of the cases in the present study could have taken antibiotics by their own judgment to control the symptoms. Because antibiotic resistance is one of the biggest threats to global health [24, 25] and diarrhea is also a common illness among the participants, all students studying abroad, irrespective of destination, need to know about prevention measures and early self-treatment of diarrhea [6].

We showed that teenage students, males and having overseas travel experience were associated with an elevated risk of diarrhea in intermediate-risk countries. Younger adult travelers may be at an elevated risk for diarrhea because they tend to worry little about the source and type of their food, and ingest a larger amount of food and thus, pathogens [5, 6, 11]. Moreover, a past study found that one additional year of age reduced risk by 1% [26]. Overseas travel experience is assumed to be an indicator of preexisting immunity to the agents causing diarrhea [6, 11–15], and sex has consistently been suggested as not associated with diarrhea [5, 6, 11]. Indeed, this appears to be inconsistent with the present finding, but it would be plausible that males and those with overseas travel experience tend to practice risky behaviors, e.g., selecting unsafe food and beverages. Further studies need to investigate whether the association of sex and overseas travel experience with diarrhea would vary by food and beverage selections while abroad.

By contrast, study major was not associated with diarrhea in either of the destination's risk categories, partly because of the small number of cases in healthcare majors. This was inconsistent with a past study which reported that healthcare major was associated with an elevated risk of diarrhea, although healthcare major appeared to be correlated with destination [8]. Moreover, the 2017 academic year was associated with an elevated risk in low-risk countries, and other characteristics were not of statistical significance. Indeed, an event resulting in an elevated risk of diarrhea occurred in low-risk countries, but the small number of cases hindered stratified analysis by destination and academic year. A larger sample size will provide further insight for the associations.

The present study has several limitations. First, the destinations included only popular study abroad countries in Asia, except for Philippines. Therefore, our findings do not seem to be applicable to university students studying abroad in intermediate-risk countries located in other regions. Second, we retrospectively collected information about diarrheal episodes one to two months after their return to Japan, thus the participants could not precisely remember the onset date, the frequency of bowel movement, symptoms, and coping strategies during the episodes. Third, 163 participants in intermediate-risk countries were censored as they spent only six days at the destination, leading to underestimation of their first 2-week diarrhea risk. To examine the robustness of the present findings, we conducted two sensitivity analyses and found no substantial changes. Fourth, this study was conducted in the spring or summer. The seasonal component may in part account for the results based on the diarrhea risk in non-tropic (low-risk) countries. Finally, we cannot rule out the possibility of residual confounding from certain genetic factors (e.g., polymorphisms in the interleukin-8 gene and lactoferrin gene) [5]. Although we conducted stratified analyses by the destination's risk categories, the results should be cautiously interpreted because students who were genetically susceptible to diarrhea may avoid programs in intermediate-risk countries.

## Conclusions

Diarrhea was a common illness among Japanese university students studying abroad, which is comparable with general travelers, e.g., tourists. Younger students, males and those with overseas travel experience need to be cautious about the risk of diarrhea, specifically in intermediate-risk countries. To lower their risk, they should avoid high risk behaviors, e.g., through careful choices of safe food and beverages.

## Supporting information

**S1 Table. Sensitivity analyses of personal characteristics associated with diarrhea, by the destination's risk categories.**
(DOC)

## Acknowledgments

We would like to thank Dr. Takao Inamori and Dr. Mayumi Ono for their cooperation in data collection. We also thank Ms. Yumiko Miyano and Ms. Shino Ishikawa for their valuable support.

## Author Contributions

**Conceptualization:** Michiyo Yamakawa, Toshihide Tsuda.

**Data curation:** Michiyo Yamakawa, Toshihide Tsuda, Keiko Wada, Chisato Nagata, Etsuji Suzuki.

**Formal analysis:** Michiyo Yamakawa, Keiko Wada, Chisato Nagata.

**Funding acquisition:** Michiyo Yamakawa.

**Investigation:** Michiyo Yamakawa, Toshihide Tsuda.

**Methodology:** Michiyo Yamakawa, Toshihide Tsuda, Etsuji Suzuki.

**Project administration:** Michiyo Yamakawa, Toshihide Tsuda.

**Supervision:** Etsuji Suzuki.

**Writing – original draft:** Michiyo Yamakawa.

**Writing – review & editing:** Michiyo Yamakawa, Toshihide Tsuda, Keiko Wada, Chisato Nagata, Etsuji Suzuki.

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
