## [Decision Letter · Decision Letter 0]

2 Sep 2022

PONE-D-22-18551Diarrhea and related personal characteristics among Japanese university students studying abroad in intermediate- and low-risk countriesPLOS ONE

Dear Dr. Yamakawa,

Thank you for submitting your manuscript to PLOS ONE. After careful consideration, we feel that it has merit but does not fully meet PLOS ONE’s publication criteria as it currently stands. Therefore, we invite you to submit a revised version of the manuscript that addresses the points raised during the review process.

Please note that we have only been able to secure a single reviewer to assess your manuscript. We are issuing a decision on your manuscript at this point to prevent further delays in the evaluation of your manuscript. Please be aware that the editor who handles your revised manuscript might find it necessary to invite additional reviewers to assess this work once the revised manuscript is submitted. However, we will aim to proceed on the basis of this single review if possible. 

 Please submit your revised manuscript by Oct 17 2022 11:59PM. If you will need more time than this to complete your revisions, please reply to this message or contact the journal office at plosone@plos.org. Please include the following items when submitting your revised manuscript:A rebuttal letter that responds to each point raised by the academic editor and reviewer(s). You should upload this letter as a separate file labeled 'Response to Reviewers'.A marked-up copy of your manuscript that highlights changes made to the original version. You should upload this as a separate file labeled 'Revised Manuscript with Track Changes'.An unmarked version of your revised paper without tracked changes. You should upload this as a separate file labeled 'Manuscript'.

We look forward to receiving your revised manuscript.

Kind regards,

Alice Coles-Aldridge

Editorial Office

PLOS ONE

Journal Requirements:

"This work was supported by JSPS KAKENHI [Grant Number JP17K02111]."

Reviewers' comments:

Reviewer's Responses to Questions

**Comments to the Author**

1. Is the manuscript technically sound, and do the data support the conclusions?

Reviewer #1: Yes

2. Has the statistical analysis been performed appropriately and rigorously? 

Reviewer #1: Yes

3. Have the authors made all data underlying the findings in their manuscript fully available?

Reviewer #1: Yes

4. Is the manuscript presented in an intelligible fashion and written in standard English?

Reviewer #1: Yes

5. Review Comments to the Author

Reviewer #1: A very nice paper on TD among students studying abroad. A very enjoyable read, robustly analyzed, and very clearly written.

A few comments for consideration:

1. Vibrio cholerae should have the species name lowercase

2. For this statement in the discussion, I think the authors mean to adjust their plans: "Another possible reason is that a longer period of stay made it easier to adjust the schedules."

3. In the limitations and the discussion paragraphs the authors also need to note that this study was done in the spring/summer and there may be a component of seasonality among non-tropics (low risk) destinations that may account for some of their risk-based findings.

4. The conclusions should be broadened. What exactly do the authors think needs to happen for these groups to LOWER their risk? In other words, what is the public health message to these groups? Simply restating results in the conclusion isn't very helpful here.

5. In the methods the authors need to state how data was managed (i.e., what software was used): SAS? R? Statistical packages should also be mentioned.

6. PLOS authors have the option to publish the peer review history of their article (what does this mean?). If published, this will include your full peer review and any attached files.

Reviewer #1: No

---

## [Author Response · Author response to Decision Letter 0]

11 Oct 2022

October 4, 2022

Dr. Alice Coles-Aldridge

Editorial Office

PLOS ONE

Dear Dr. Aldridge,

 On behalf of the authors, I am pleased to submit our revised manuscript entitled “Diarrhea and related personal characteristics among Japanese university students studying abroad in intermediate- and low-risk countries” for possible publication as an original article in PLOS ONE. The revised manuscript consists of 225 words for abstract 2,489 words for main text, 4 tables, and 1 online material (1 supporting information file).

 Following the reviewer’s comments, we carefully revised our manuscript, and our point-by-point responses to the comments are attached.

 

Responses to Academic Editor

We thank the academic editor for this opportunity to revise our manuscript. We carefully revised the manuscript following the reviewer #1’s comments. First, we added the sentences in the limitation of Discussion section that because this study was conducted in the spring or summer, seasonal component may in part account for the risk-based findings in non-tropic countries. Next, to lower the risk of diarrhea in teenagers, males, and those with overseas travel experience, who were suggested to be high risk, we added public health messages for the groups in the Conclusion section. In addition, we modified the manuscript following the journal style requirements, provided additional information regarding participant consent and funding statement, and included captions for our supporting information file. As a result of this revision, we believe that our manuscript has greatly improved. 

Responses to comments from Reviewer #1

A very nice paper on TD among students studying abroad. A very enjoyable read, robustly analyzed, and very clearly written.

We thank the reviewer for his/her constructive comments and carefully revised the manuscript following the comments.

1. Vibrio cholerae should have the species name lowercase.

Response:

Thank you for your pointing out. We corrected the species name in lower case as follows:

“Considering the immunity against norovirus [14] and Vibrio cholerae [15], ABO blood group was dichotomized as ‘O’ or other types including unknown blood type.” (page 5, line 83)

2. For this statement in the discussion, I think the authors mean to adjust their plans: "Another possible reason is that a longer period of stay made it easier to adjust the schedules."

Response:

Thank you for your pointing out. We replaced the word “schedules” to “plans” in the sentence as follows: 

“Another possible reason is that a longer period of stay made it easier to adjust their plans.” (page 13, lines 217 to 218)

3. In the limitations and the discussion paragraphs the authors also need to note that this study was done in the spring/summer and there may be a component of seasonality among non-tropics (low risk) destinations that may account for some of their risk-based findings.

Response:

We agree. While acknowledging the potential impact of seasonal component on the risk-based findings in non-tropic (low-risk) countries, we thus added the sentences in the limitation of Discussion section as follows: 

“Fourth, this study was conducted in the spring or summer. The seasonal component may in part account for the results based on the diarrhea risk in non-tropic (low-risk) countries.” (page 14, lines 259 to 261)

Results

4. The conclusions should be broadened. What exactly do the authors think needs to happen for these groups to LOWER their risk? In other words, what is the public health message to these groups? Simply restating results in the conclusion isn't very helpful here.

Response:

We agree. To lower the risk of diarrhea in teenagers, males, and those with overseas travel experiences, who were suggested to be high risk, public health messages for the groups are necessary. We thus modified the sentences in the Conclusion section as follows:

“Diarrhea was a common illness among Japanese university students studying abroad, which is comparable with general travelers, e.g., tourists. Younger students, males and those with overseas travel experience need to be cautious about the risk of diarrhea, specifically in intermediate-risk countries. To lower the risk, they should avoid high risk behaviors, e.g., through careful choices of safe food and beverages.” (page 15, lines 268 to 272)

5. In the methods the authors need to state how data was managed (i.e., what software was used): SAS? R? Statistical packages should also be mentioned.

Response:

Thank you for your pointing out. We added the information about the statistical package we used as follows:

“Stata SE statistical software (version 16.1; StataCorp, College Station, TX, USA) was used for all the analyses.” (page 7, lines 129 to 130)

 

We believe that our paper has improved as a result of the reviewer’s helpful suggestions. We thank you for the opportunity of re-submission. We hope that, with these modifications, our paper can now be accepted for publication. We look forward to hearing from you.

Sincerely yours,

Michiyo Yamakawa, PhD, on behalf of all authors

---

## [Decision Letter · Decision Letter 1]

6 Dec 2022

Diarrhea and related personal characteristics among Japanese university students studying abroad in intermediate- and low-risk countries

PONE-D-22-18551R1

Dear Dr. Yamakawa,

We’re pleased to inform you that your manuscript has been judged scientifically suitable for publication and will be formally accepted for publication once it meets all outstanding technical requirements.

Kind regards,

Nadim Sharif, M.Sc.

Academic Editor

PLOS ONE

Reviewers' comments:

Reviewer's Responses to Questions

**Comments to the Author**

1. If the authors have adequately addressed your comments raised in a previous round of review and you feel that this manuscript is now acceptable for publication, you may indicate that here to bypass the “Comments to the Author” section, enter your conflict of interest statement in the “Confidential to Editor” section, and submit your "Accept" recommendation.

Reviewer #1: All comments have been addressed

Reviewer #2: (No Response)

2. Is the manuscript technically sound, and do the data support the conclusions?

Reviewer #1: Yes

Reviewer #2: Yes

3. Has the statistical analysis been performed appropriately and rigorously? 

Reviewer #1: Yes

Reviewer #2: Yes

4. Have the authors made all data underlying the findings in their manuscript fully available?

Reviewer #1: Yes

Reviewer #2: Yes

5. Is the manuscript presented in an intelligible fashion and written in standard English?

Reviewer #1: Yes

Reviewer #2: Yes

6. Review Comments to the Author

Reviewer #1: Thank you for addressing my concerns. I have no further comments. I appreciate the author's diligence in addressing them.

Reviewer #2: This manuscripts highlights the problem of diarrheal illnesses associated with student travel to LMICs. This is of concern because study abroad opportunities in LMICs are increasing in popularity with college students. The authors collected data from several hundred students and their analysis was sound. The evidence supports their conclusions that travelers' diarrhea is a significant risk to students traveling to LMICs and that students should be diligent in their personal protection to decrease the risk of illness. Any issues I had with this paper were addressed by the other reviewer and the authors have satisfactorily addressed the comments.

7. PLOS authors have the option to publish the peer review history of their article (what does this mean?). If published, this will include your full peer review and any attached files.

Reviewer #1: No

Reviewer #2: **Yes: **Kelli L. Barr

---

## [Editor Report · Acceptance letter]

13 Dec 2022

PONE-D-22-18551R1 

Diarrhea and related personal characteristics among Japanese university students studying abroad in intermediate- and low-risk countries 

Dear Dr. Yamakawa:

I'm pleased to inform you that your manuscript has been deemed suitable for publication in PLOS ONE. Congratulations! Your manuscript is now with our production department. 

Kind regards, 

on behalf of

Dr. Nadim Sharif 

Academic Editor

PLOS ONE